# Performance Evaluation of Fixed-Point Continuous Monitoring Systems: Influence of Averaging Time in Complex Emission Environments

**DOI:** 10.3390/s25092801

**Published:** 2025-04-29

**Authors:** David Ball, Nathan Eichenlaub, Ali Lashgari

**Affiliations:** Project Canary, Denver, CO 80202, USA; david.ball@projectcanary.com (D.B.); nathan.eichenlaub@projectcanary.com (N.E.)

**Keywords:** methane emissions, leak detection and repair, emissions quantification, fixed-point sensors, continuous monitoring systems, source-level emissions, site-level emissions

## Abstract

Quantifying methane emissions from facilities with complex emissions profiles can present a substantial challenge. Real-world emission scenarios can involve dynamic operational background emissions and temporally overlapping asynchronous emission events with varying rates from multiple sources. Previous studies have involved simpler testing setups, often with synchronous emission sources and constant rates. This work is among the first to assess the performance of continuous monitoring systems (CMSs) under dynamic, overlapping emission scenarios with time-varying baselines. The data were collected as part of a novel single-blind controlled release study, where release sources and emission rates are not disclosed during the testing period. Several error metrics are defined and evaluated across a range of relevant averaging times, demonstrating that despite significant error variance in short-duration estimates, the low bias of the system results in substantially improved emission estimates when aggregated to longer timescales. Over the 4-week duration of this study, 700 kg of methane was released by the testing center, while the estimated quantity shows a final mass of 673 kg, an underestimation by 27 kg (4%). These results demonstrate that advanced CMSs can accurately quantify cumulative site-level emissions in complex scenarios, highlighting their potential for enhanced future emissions monitoring and regulatory applications in the oil and gas sector.

## 1. Introduction

Fixed-point continuous monitoring systems (CMSs) have been deployed in oil and natural gas production facilities over the past few years, primarily to provide a continuous stream of data related to site-level methane emission, allowing for the detection of anomalous emission events, source localization, and emission rate quantification. CMSs can also complement other forms of methane monitoring by providing data-driven site-level insights regarding emissions event duration and frequency [1], and also provide complementary meteorological data to other measurement modalities that do not have in situ sensors (e.g., aerial flyovers or satellite overpasses).

The growing demand for continuous measurement of methane, given its clear benefits related to operational improvement and regulatory compliance, requires proper performance evaluations to ensure accurate performance under realistic and complex emission scenarios, including multi-source asynchronous releases and time-varying baseline emissions. Such complex emission environments are often under-represented in previous technology evaluation studies. This motivates the need to develop new testing protocols to more closely mimic actual operational conditions. In addition to the testing efforts, it is essential to perform in-depth analyses of the system performance to understand capabilities and limitations of each technology.

The accurate and reliable performance of CMSs is crucial in real-world applications related to emissions inventory development, regulatory compliance, and operational improvement. For instance, the EPA’s OOOOb regulation permits the use of CMSs as an alternative test method for emissions compliance, provided specific criteria are met. These include (i) “… continuous monitoring means the ability of a methane monitoring system to determine and record a valid methane mass emissions rate or equivalent of affected facilities at least once for every 12-hour block”; and (ii) “the 90-day rolling average action-level is 1.6 kg/h (3.6 lb/h) of methane over the site-specific baseline emissions” and “the 7-day rolling average action level is 21 kg/h (46 lb/h) of methane over the site-specific baseline emissions.” Building upon these regulatory developments, reliable and accurate performance of the CMSs becomes even more critical to meet the requirements outlined by the EPA.

Various ambient methane measurement technologies exist for fixed-point continuous methane monitoring systems, spanning a wide range of detection modalities. Each of these sensing options have their own unique strengths and limitations. These technologies offer adaptability to various environmental conditions and application requirements. For instance, metal oxide (MOx) sensors provide cost-effective, broad-range concentration measurements. However, their technological limitations often restrict the utilization of these sensors to anomaly detection [1,2]. In contrast, tunable diode laser absorption spectroscopy (TDLAS) sensors offer high precision and selectivity in return for a higher sensor price point. The selection of the measurement technology depends on factors such as project objectives, sensitivity requirements, operational environment, and cost considerations [3,4]. Examples of other ground-based continuous methane measurement methods include fixed optical gas imaging camera systems with quantitative capabilities, as well as path-integrated methane measurement technologies that measure concentrations across short-range distances (e.g., <200 m) or over kilometer-scale areas. The accuracy of emission rate quantification using these systems may vary significantly depending on the technology and solution provider [5,6,7].

Properly deployed CMSs can provide timely alerts of potential site-level methane release events that could lead to elevated concentrations using a wide range of algorithms, from static ambient concentration thresholds to sophisticated machine learning techniques [8]. For the first few years of the at-scale deployment of CMSs, anomalous event detection was considered the primary application of these systems. However, with advances in emissions dispersion modeling and associated rate inversion, CMSs have demonstrated potential beyond emission event detection. Enhanced emission modeling can result in reliable source localization and emission quantification, which can significantly augment the actionable insights derived from these systems [9].

Fixed-point sensors provide ambient methane concentration measurements, often in parts-per-million (ppm), at a known location at a relatively high temporal frequency (typically at least one measurement is reported every minute). The raw data from CMSs often consist of a set of methane concentration measurements in several sensor locations, plus meteorological measurement data collected using on-site anemometers. To infer the flux rate at the source location(s) (mass of pollutant emitted per unit of time), quantification algorithms need to translate the CMS raw data into the mass of pollutant emitted per unit of time. This is often achieved by combining the application of forward dispersion models and inversion frameworks [10].

Forward dispersion models and inversion frameworks are often employed for the estimation of source emissions rate based on ambient concentration measurements. Forward dispersion models simulate the atmospheric transport of pollutants from any given source to receptors (e.g., sensor locations), factoring in meteorological parameters such as wind speed, wind direction, and atmospheric stability. In other words, a forward dispersion model simulates concentration enhancements at a given location, resulting from a known release rate from a given source location [4,11,12,13,14,15,16,17,18,19,20]. Subsequently, inversion models use mathematical optimization to determine source flux rates that produce simulated concentrations that align as closely as possible to the actual measurements. Inversion frameworks try to leverage simulated concentrations at the sensor locations and knowledge of forward dispersion patterns to solve an optimization problem, where the objective is to find a combination of source emission rates generating simulated concentrations that best fit the observed concentrations [8]. A detailed discussion on the performance of various forward dispersion models and inversion frameworks can be found in [21].

Controlled release studies are invaluable in helping to improve the capabilities of CMSs technologies and evaluate their performance. These studies provide large volumes of high-quality “ground truth” data that enable technology developers to drive innovation and improve their algorithms. Recent studies suggest that the performance of CMSs solutions has improved through continuous, rigorous testing using a standardized protocol [7]. Cheptonui et al. [7] indicated a positive correlation between repeated testing (frequent participation in controlled release testing studies) and improvement in the overall performance of solutions. The bulk of these improvements are realized via improved algorithmic workflows, from data preprocessing and cleaning, to more accurate dispersion modeling, and finally implementing more sophisticated inverse solvers; the hardware being tested tends to stay the same year-over-year.

More specifically, single-blind controlled release studies use several pre-defined metrics to assess the performance of CMSs solutions. These evaluations encapsulate both the emissions measurement (hardware) and analytics (algorithms) related to emissions detection, localization, and quantification (DLQ). In single-blind studies, known quantities of natural gas are emitted from one or several release points within the study site. Each participating technology submits a summary of its DLQ results without prior knowledge of emissions rates, release points, or the timing of emission events. Submitted results are then compared against ground-truth data to assess how well each technology performed during the study [22]. Note that controlled release studies are often designed to determine the combined uncertainty resulting from measurement (hardware) and data analysis (algorithms). A recent study [1] focused on processing concentration data from three different measurement technologies using a standalone, open-source algorithm. They also compared the results for their model to the emission rates from solution providers. Note that [1] focused on evaluating the performance of an open-source quantification algorithm using several sets of data, as opposed to employing a ground-truth dataset to perform comparative analysis of the uncertainty associated with various models. To the best of our knowledge, there is no testing campaign that has been undertaken to disentangle these two sources of uncertainty and focus on model performance evaluation based on the same gold-standard measurement data by, for example, providing the participants with the same measurement data and focusing solely on the performance of different algorithms applied to the same underlying data.

In terms of facility complexity, the layout of controlled release sites can be simple (where only one or a few release points are included in the experiments and no obstacles or complex terrain is present), moderate (such as controlled testing facilities specifically designed to simulated operational emissions), or complex (such as actual operational oil and gas facilities). Other factors, such as complex terrain or the presence of obstacles, may contribute to the complexity of the testing facility. In addition, controlled releases can also vary in terms of the complexity associated with emission scenarios. First, a controlled release experiment may include a single emissions event or multiple overlapping events (i.e., multiple active release points). Second, when the emission scenario includes multiple overlapping events, those events can start and end together or asynchronously. Third, emission events may consist of steady-state or time-varying emission rates. Fourth, emission events may be designed to occur in the absence of simulated baseline emission, or alternatively, a simulated baseline emission level (steady-state or fluctuating baseline) may be present. Fifth, to add to this complexity, fluctuations in the baseline emissions may be designed to be comparable to the emission event rates. Sixth, emission scenarios may be designed with various durations and magnitudes, ranging from short-duration, small events to long-lasting events with high emission rates. Seventh, release points may be underground (to simulate pipelines), on the surface, or significantly elevated (for instance, representative of tanks or flare stacks). Eighth, offsite emission sources can be included in the design of emission scenarios. Lastly, in the case of non-oil and gas controlled releases, area sources may be considered in designing emission scenarios (e.g., to simulate landfill or underground pipeline emissions).

Examples of methane controlled release studies for CMSs solutions include testing under the Advancing the Development of Emission Detection (ADED) program [22], funded by the US Department of Energy’s National Energy Technology Laboratory (NETL), administered by the Colorado State University’s Methane Emissions Technology Evaluation Center (METEC) in its Fort Collins facility [6,7,23] and during field campaigns [24,25,26]; studies performed at the TotalEnergies Anomaly Detection Initiative (TADI) testing facility [27,28,29]; Stanford University’s controlled release study in an experimental field site in Arizona [5]; Highwood Emissions Management testing in Alberta [30]; and Alberta Innovates technology-specific controlled release testing studies [31]. Note that some of these studies include simple release scenarios with only one release point [30], while others may range from moderate complexity, with multiple release points with simplified release scenarios (e.g., steady-state, synchronous events during each experiment) [6,7,23], or complex release scenarios conducted in actual operational setups [24,26].

Controlled release testing studies conducted by METEC from 2022 to 2024 [6,7,23] are known as the most comprehensive single-blind CMSs controlled release studies. The first iteration of the ADED protocol [22] was employed during these studies. This protocol is comprised of temporally distinct “experiments” at the METEC facility, each of which has between 1 and 5 synchronous release events (turned on and off at the start and end of each unique experiment). During the most recent (2024) campaign, experiment durations ranged between 15 min and 8 h. Emission rates for each source during an individual experiment were held constant, with individual source rates ranging from 0.081 to 6.75 kg/h. In that study, the experiments were designed such that only one release was active per equipment group at the METEC facility. Emission scenarios were designed in the absence of an artificial baseline emission or off-site sources. The results of the 2024 ADED study are publicly available on the METEC website [32].

While research efforts have primarily concentrated on evaluating the accuracy of fixed-point CMSs-based quantification for steady-state emission releases, studies have recently started to focus on the more complex scenario of dynamic and asynchronous emissions, which are common in operational facilities. Simpler event-based emission patterns used in previous studies characterized by distinct “experiments” with multiple synchronized release points do not appropriately mimic the complex emissions expected at operational facilities, and therefor do not fully evaluate the efficacy of the solutions being tested in terms of their performance in the field. As such, there is a clear need for advanced testing protocols that are capable of more closely simulating real-world emissions patterns and evaluating the performance of technologies under these more complex situations.

CSU’s METEC has recently upgraded its facility in Fort Collins and published a revised ADED testing protocol aiming to better align controlled release testing with emission profiles of real-world operating oil and gas facilities [33]. This upgraded METEC 2.0 facility will enhance testing capabilities by adding new equipment, expanding release point options, and improving underground controlled release testing capabilities. In addition to these physical upgrades, the future ADED testing will include more complex emission scenarios, including fluctuating baseline emissions, asynchronous releases, and time-varying release rates. More information regarding the METEC 2.0 Controlled Test Protocol can be found on the ADED 2.0 webpage (accessed on 26 April 2025): https://metec.colostate.edu/aded-2-0/.

This study evaluates the performance of emissions quantification using fixed-point continuous monitoring systems under complex single-blind testing conditions that more closely mimic real-world operational emission scenarios. More specifically, by comparing the actual release rates to the estimated rates, this study investigates the impact of averaging time on the uncertainties associated with emissions quantification. We take a deep dive into the application of short-term and long-term emission rate averaging approaches and study the root causes of emission source misattribution in a few select scenarios. Lastly, the application of anomalous emissions alerting above baseline is investigated. To the best of our knowledge, this study is the first of its kind concerning the performance evaluation of a fixed-point continuous monitoring system under single-blind testing conditions involving complex, multi-source emission scenarios, including fluctuating baseline emissions, asynchronous releases, and time-varying release rates. Although the data are collected using a specific fixed-point CMSs solution (employing TDLAS sensors) and a solution-specific quantification method is employed to derive emission rates, we still expect that some of the insights discussed in this study hold for many analogous technologies. More specifically, the insights related to the impact of averaging time on various performance metrics are expected to hold valid for systems that demonstrate low bias but significant error variance, of which there may be several (see, e.g., [7]). While specific numbers and convergence features will be dependent on specific algorithms and processing techniques, the general trends are expected to be similar. Furthermore, conclusions related to the capability of point-in-space gas analyzers to disentangle emissions between upwind sources during periods of low wind variability are also generally expected to hold for analogous technologies.

## 2. Data and Method

In August and September of 2024, METEC conducted a 28-day trial study intended to more accurately mimic emissions at operational oil and gas facilities via the inclusion of a noisy and time-varying baseline layered with multi-source asynchronous releases of various sizes. Project Canary participated in this single-blind controlled release study. It should be noted that the description here represents the authors’ best understanding of the controlled release study experiments and does not come directly from METEC.

The revised ADED testing protocol aims to replicate the emission characteristics of operational facilities, incorporating a stochastic, time-dependent baseline with significant high-frequency spectral power originating from diverse spatial locations. This baseline is intended to simulate operational emissions, such as those from pneumatic devices and compressor slip. Following a one-week baseline emission period, controlled releases of varying magnitudes and durations were introduced. These release events may exhibit partial or complete temporal overlap. At any given time, the total rate can be computed as the sum over all individual releases. This quantity will be often referred to as the “site-level” or “source-integrated” rate, and represents the total emission rate from the facility. During the baseline week, the site-level emission rate was 0.4 kg/h, with a maximum site-level rate of 4.9 kg/h. After this period, the following three weeks had an average site-level emission rate of 1.2 kg/h and a maximum value of 9.3 kg/h. The minimum site-level emission rate for both the baseline period and the following three weeks was 0 kg/h (i.e., there are periods of time when there are no emissions from the facility). The standard deviation of site-level rates during the baseline period was 0.5 kg/h, and 1.5 kg/h for the following three weeks. In other words, the first week of “operational” or “baseline” emissions is characterized by a relatively small rate without too much variance (smaller mean, maximum, and standard deviation), while the following three weeks of testing include larger and more highly varying emissions, resulting in a higher mean and standard deviation in emission rate as compared to the baseline week.

This study utilizes data collected using the Canary X integrated device, which combines an enclosed-cavity TDLAS gas analyzer with an optional RM Young 2-dimensional ultrasonic anemometer. The Canary X, an IoT-enabled monitor, leverages high-sensitivity methane concentration measurements and meteorological measurements for methane emissions quantification. The methane sensor features a 0.4 ppm sensitivity, ±2% accuracy, and ≤0.125 ppm precision (60-s averaging). While both the anemometer and gas analyzers perform high-frequency (>1 Hz) sampling of their respective measurements, these are aggregated to every minute via an average, and only these minute-averaged quantities are reported to the cloud-based server and used in the following quantification algorithms. In order to visually illustrate the data collected during this trial period and the associated layout of sources with respect to sensors, Figure 1 shows the facility layout in the left panel, with source groups indicated with solid color-coded boxes and sensor locations indicated with colored x’s. The right panel shows the concentration measurements from each sensor during the 28 days of data collection, with the colors and positional labels (e.g., “SW”) corresponding to the x’s in the left panel. The wind rose associated with these 28 days is shown in Figure 2, showing that the dominant wind direction is along the SE/NW axis, but has significant variability, with every wind direction bin having a substantial number of associated measurements.

A comprehensive analysis comparing various quantification methodologies, including different combinations of forward dispersion models and inverse frameworks, was discussed in a prior publication [21]. The focus of this study lies in evaluating the system’s performance under complex, real-world emission scenarios and assessing the impact of averaging time on emission rate estimation. Therefore, the core insights derived in this work, particularly regarding the trade-offs between temporal resolution and error reduction, will generally apply to fixed-point CMSs, regardless of the specific algorithm used. By concentrating on the examination of the system’s overall performance and operational implications, this study aims to improve the understanding of methane measurement and quantification using fixed-point CMSs, building upon our prior work on quantification methodologies [21].

The selection of the proper performance evaluation metrics for CMSs applications depends on the use case and the objective of the monitoring program. This study investigates the performance of one such system across a broad range of evaluative metrics, beginning with a direct comparison of 15 min source emission rate estimates against ground-truth values. Subsequently, an analogous analysis is presented, focusing on extended-period time-averaged emission rates, which are generally more robust for informing actionable insights. This approach is grounded in findings from previous studies, which consistently indicate a high degree of variance in instantaneous quantification errors, but a low amount of bias, suggesting that longer-term (i.e., time integrated or averaged) estimates are generally more robust [1,7]. Next, cumulative mass emission estimates are compared to the actual cumulative gas release volumes to provide a comprehensive understanding of the system’s performance over time. This analysis is performed both at the site-level and also per source-group, to assess the efficacy of the system at accounting for the total mass being emitted by the facility as well as its ability to properly allocate that mass to spatially clustered pieces of equipment. A few examples are then presented along with an investigation of the root cause of source misattribution. Finally, to assess the effectiveness of these systems in identifying significant deviations from normal operating conditions (represented by the first week of emissions during the testing), a threshold-based analysis is employed to evaluate the system’s capacity to detect and alert anomalous emissions exceeding established operational baselines.

## 3. Results

In this section, we present evaluations of the system’s output with respect to several use cases: Section 3.1 shows a direct comparison of raw source rate estimates to ground-truth values, while Section 3.2 presents a similar analysis on time-averaged rates in order to more directly analyze an aggregate output of the system that is generally more useful in providing actionable information. Section 3.3 analyzes the cumulative mass emission curves at a facility level as well as per equipment group, and provides a brief exploration of the underlying causes of source misattribution. Finally, Section 3.4 applies thresholds to time-averaged rates to evaluate the ability of the system to detect and alert of anomalous emissions above an operational baseline.

All of the metrics in the following sections are computed over the entire 28 days of data: there was no system downtime or time periods that were excised for the purpose of the analysis. The native temporal resolution of the quantification system is 15 min (i.e., a quantification estimate is produced every 15 min for each equipment group), resulting in a total of 2687 quantification estimates at the output resolution of the system for each equipment group. This is the same output frequency that is used in all field deployments of this system.

### 3.1. Instantaneous Rate Error

In this section, we compare the rate estimates generated every 15 min as an output of the quantification algorithm to the 15 min aggregate ground-truth release rates as reported by the testing center. Figure 3 shows stacked bar plots of the ground-truth (top) and estimated (bottom) emissions for each equipment group (distinguished by color) across the 4-week single-blind testing period. In general, there is significantly more variability in the estimated rates due to the inherent uncertainties associated with short-duration rate estimates, as has been thoroughly documented in the previous literature [1]. Despite this high-frequency noise in the estimates, there is reasonably good correspondence between the estimated equipment group-specific rates and the ground-truth release rates. Figure 3 indicates that the dominant colors generally align between the two plots and the overall magnitude of individual releases (as well as source-integrated whole site emission rate) shows reasonable agreement.

To more quantitatively assess the site-level rate error associated with the results shown in Figure 3, emission rate errors are determined by comparing each 15 min site-level rate estimate to the ground-truth values. Figure 4 shows the error distribution of source-integrated rate estimates on the left and a parity plot of the 15 min site-level rate estimates and actual rates on the right. The mean error (a direct measure of the bias) is shown with a vertical dashed orange line. Note that the cumulative mass estimate error will be equal to $\bar{E}\Delta t$, where $\bar{E}$ is the mean error and Δt is the duration of the entire experiment (28 days). As is evident from the error histogram, the error distribution has a large central peak near 0 and is roughly symmetric, resulting in a near-zero mean error of −0.04 kg/h. In other words, on average, the system tends to underestimate 15 min rate estimates by 0.04 kg/h compared to the actual rates during this study. Plus and minus the mean absolute error $\bar{\left|E\right|}$ are depicted with red dotted lines. These are effectively a measure of the characteristic width of the error distribution, and represent how far off in absolute magnitude an individual rate estimate is from the actual rate, on average. In this case, individual 15 min rate estimates deviated from the actual rates by 0.66 kg/h, on average. These metrics imply a nearly zero centered error distribution with significant width relative to the typical rates employed during this study, consistent with the expectation of there being a high degree of short-term error variance, as illustrated in Figure 3.

The right panel of Figure 4 shows the parity plot of site-level rates. Here, the horizontal axis is the actual emission rate from the facility, while the vertical axis corresponds to the estimated emission rates. The dashed black line depicts the parity relation (x=y) and the solid orange line shows the best linear fit to these data. The linear fit has a slope of 0.82, indicating that the system has a tendency to underestimate instantaneous rates, which is consistent with the mean error shown in the left panel of the same figure being negative. The $R^{2}$ of the linear fit is 0.38, indicating substantial variance in the distribution about the line of best fit, consistent with the relatively high value of $\bar{\left|E\right|}$. As demonstrated here, the instantaneous rate error has quite a bit of scatter to it (as evidenced by the relatively high $\bar{\left|E\right|}$ and low $R^{2}$). As a result, making decisions based on short-duration estimates (e.g., deploying a field team to perform leak detection and repair inspection visits using optical gas imaging cameras or inspecting a piece of equipment for an underlying issue) may yield suboptimal results (i.e., false positive detections). The near-zero bias, however, indicates that alerts based on longer integration time periods should be far more reliable. These results highlight the importance of longer time-integration for deriving actionable insights from CMSs-inferred quantification.

### 3.2. Time-Averaged Rate Error

As demonstrated in Section 3.1, short-term, 15 min rate estimates (associated with the native output temporal resolution of the quantification algorithm that is deployed at scale) show a significant amount of variance in the error distribution, and as such, should be interpreted with caution if being used in a decision-making process. Despite short-term variability, the near-zero bias suggests that averaging over a longer time period, long enough to reduce error variance but at the same time not so long that relevant emission variations are smoothed over, can provide operators with more reliable emission rate estimates that lead to operational insights and better-informed decision making. For this purpose, we adopt an averaging time of 12 h. This particular choice of averaging time is informed by the EPA OOOOb continuous monitoring requirement, indicating, among other things, that “… *continuous monitoring means the ability of a methane monitoring system to determine and record a valid methane mass emissions rate or equivalent of affected facilities at least once for every 12 hour block*” [34]. In addition, this longer time window represents many multiples (specifically 48) of the system’s output frequency, meaning there is ample opportunity for the highly varying instantaneous errors to average out. It is also short enough to represent a daily operational emission profile: day-to-day variations in emissions will be meaningfully captured and can be used to directly probe the impact of certain activities or known emission events on what is effectively a “working day average”. For the purposes of this analysis, temporal aggregates are taken in discrete non-overlapping 12 h blocks over which averages of both the ground-truth rates and estimates are computed.

Figure 5 shows an analogous stacked line plot to Figure 3, but aggregated to 12 h mean rate values for each equipment group on the facility. The improvement via a visual comparison of equipment-specific source rates is immediately obvious: there is significantly less variation in the estimated source rates about the actual rates. While the 12 h aggregates generally show good agreement between the actual and estimated values in terms of the total site rate and the dominant emitters, it is worth noting that the estimated quantities show signs of overproducing nonzero sources. For example, during the large peak in emissions that occurs dominantly from the 4W and 5S equipment groups on 27 August, there is very little contribution to the total site flux from any other equipment group. The estimated quantity, however, shows significant nonzero emissions from the 4T and 4S groups. This highlights an important feature of the quantification algorithm that is employed here: it is prone to source misattribution and has the tendency to overpredict the number of active emitters in a given time period. While some source misattribution is evident, the dominant emitters are properly identified in all cases, and a smaller portion of flux is erroneously assigned to inactive emission sources. This finding is consistent with [21], which found that for constant-rate emissions inference, quantification algorithms had the tendency to produce a significant number of “False Positive” emitters relative to ground truth by erroneously assigning a portion of the emissions to inactive sources.

In order to more quantitatively demonstrate what is visually evident, the number of sources with rates above 0.05 kg/h are computed at every timestep from the 12 h aggregates. The timeseries of the number of active emitters (both estimated and actual) are shown in Figure 6, and the mean values are shown in the legend. During this time period, the estimated number of emitters is larger than the actual number of emitters 62% of the time, equal to the actual number of emitters 24% of the time, and smaller than the actual number of emitters 14% of the time. The mean actual number of emitters over this time period is 1.9, while the estimated quantity is 2.9. In other words, on average, the quantification estimates overestimate the number of significant emission sources by 1.

This highlights an important limitation of the existing quantification algorithms that use fixed-point monitoring data: while the estimation of site-level emission rate and identification of dominant emitters is promising, the small attribution of rate to non-emitting pieces of equipment should be taken into consideration when interpreting the results. In other words, during periods with elevated concentrations, the attribution of minor emission rates to secondary potential sources should be interpreted with caution.

Similar to Figure 4, the error distribution of source-integrated rate estimates and a parity plot of the 12 h averaged site-level rate estimates and actual rates are shown in Figure 7. The left panel shows that the characteristic width of the error histogram shrinks substantially, which is reflected in the $\bar{\left|E\right|}$ metric (0.34 kg/h compared to 0.66 kg/h as shown in Figure 4). In other words, the mean absolute error almost halves when going from 15 min averaged rate estimates to 12 h aggregates. The right panel of this figure shows the parity plot of 12 h aggregate release rates against the corresponding estimates. It is immediately evident that there is far less scatter about the liner of best fit, resulting in a substantially higher $R^{2}$ of 0.73 (compared to 0.38 for the 15 min averaged rate estimates).

To more thoroughly investigate the impact of averaging time on the resulting error distribution of rate estimates, we aggregate the 15 min estimates and ground-truth rates to a range of averaging times, from 15 min to 24 h. This specific range of averaging periods is limited by the duration of this trial period of the testing protocol: in principle employing 7-day and 90-day rolling averages will be relevant for application to OOOOb Continuous Monitoring requirements, for which action thresholds are defined by these longer-term rolling averages. As such, the averaging times employed are meant to be representative of timescales on which granular day-to-day emissions insights may be useful. When data from longer-duration tests are available, however, an analogous evaluation should be assessed on 7- and 90-day averages to evaluate the accuracy of technologies on these regulatory-relevant timescales.

For each aggregation time period, the mean absolute error and root-mean-squared error as well as the slope of the parity line and associated $R^{2}$ are computed. This analysis is bootstrapped over 100 randomly sampled realizations of the underlying data for every averaging period. More specifically, at a given averaging period, 75% of the individual rate estimates and associated ground-truth values are randomly sampled from the full timeseries, with replacement, representing an individual random realization of the underlying estimates and true rates, from which the relevant error statistics are computed. This process is repeated 100 times to generate a distribution of error metrics for every averaging time, from which the mean and standard deviations are reported. Note that for the largest averaging times, there are fewer underlying data points to sample from (the testing is 4 weeks, so employing a 24 h average results in 28 distinct data points), and as such, the bootstrapped variance is expected to increase as a function of averaging time. Future controlled release studies with longer testing periods and a wider range of emissions patterns and rates will improve the understanding of the error distributions associated with these longer timescales.

Figure 8 shows the mean absolute error in both absolute (left) and relative (right) units, from the native output resolution (15 min) to 1 day (24 h). For the purposes of computing the relative, or “normalized” mean absolute error (which is computed by dividing the absolute error by the actual rate at every time interval), time periods with actual rates less than 0.1 kg/h are excised to avoid divide-by-zero errors and not overly penalize irrelevantly small absolute overestimations when the true rate is close to 0. As evident in both error metrics, the magnitude of the characteristic error shows an initial dramatic decrease with increasing averaging time, dropping by a factor of approximately 2 from the 15 min values to a 4 h average. The error begins to level off beyond this, decreasing more slowly toward longer averaging times.

These typical errors and associated trends are likely due to a couple of underlying physical factors. First, quantification algorithms are rooted in a mathematical model of gas dispersion. While these models can accurately capture the general behavior, and hence, on average accurately predict concentrations, they are not expected to be **instantaneously** correct and account for site-specific small-scale turbulence at a high frequency. As such, short-duration rate estimates are expected to be prone to some error simply due to the inherent uncertainty associated with gas dispersion models. Second, in order to gain information from potential sources, the wind must be oriented in such a way that advects pollutant directly from a potential source to a sensor in the network. As a result, there can be periods of time where the system is effectively “blind” to certain sources, during which, if rates change significantly, substantial error can result until the wind shifts towards a more favorable direction.

Figure 9 shows the slope of the best-fit parity line (left panel) and associated $R^{2}$ (right panel) of the actual-vs.-estimated site-level rates across the same averaging time periods shown in Figure 8. These are computed exactly as shown in the right panels of Figure 4 and Figure 7. The slope of the parity line at the native output resolution (15 min) is 0.82, and the $R^{2}$ is 0.38. Both of these quantities show a rapid improvement as the averaging time increases to 4 h, and then level off at around 0.9 and 0.75, respectively.

Figure 8 and Figure 9 demonstrate a notable performance improvement with increasing the averaging time from 15 min to 4 h. This improvement is characterized by a considerable reduction in error metrics and a more favorable parity plot of the estimated and actual emission quantities. Extending the averaging time from 4 to 12 h continues to yield some gains in performance metrics, but the improvement starts to diminish. Beyond the 12 h averaging threshold, performance improvement is marginal, and most of the plots exhibit a plateaued behavior, indicating that further increases in averaging time provide minimal additional gains in accuracy or precision and, in turn, reduce the resolution of granular details related to emission events. Therefore, the selection of an appropriate averaging time necessitates a balance between minimizing error and maintaining the temporal resolution required to capture meaningful emission event details.

#### Dominant Emitter Identification

In addition to computing the site-level rate error as a function of averaging time, a more granular analysis of localization accuracy as a function of averaging time is presented here. For this analysis, the dominant emitter (i.e., the equipment group with the maximum rate) for every timestamp is identified in both the ground-truth rates as well as the estimated rates. If both the estimated and actual dominant emitters are the same, then this time period is flagged as a “1”, denoting that the system accurately identified the dominant equipment group with respect to emission rate at the given timestep. Otherwise, this timestep is marked as a “0”. The percentage of timestamps that the system accurately identified the dominant equipment group is then reported across a variety of averaging periods. Time periods where the actual site-level rate is less than 0.1 kg/h are excised from this analysis.

The results of this analysis are shown in Figure 10. At the native output resolution (15 min), the dominant emitter is correctly identified 79% of the time. As the averaging time increases to 4 h, this metric improves to 86%, at which point it levels off. This indicates that although there is often visually evident source misattribution, most of the time, the system was able to correctly attribute the maximum rate to the proper equipment group.

### 3.3. Cumulative Emissions Estimates

In some cases, the objective of deploying a CMS may be to estimate the cumulative site-level emissions over a long time period. For instance, when developing measurement-informed emissions inventories, the objective is to come up with accurate estimates of either source-level or site-level (depending on the application) average annualized emissions, regardless of more granular details associated with shorter-duration emission events. In this section, the cumulative site-level emissions are computed over time and compared against the ground-truth mass of gas released by the testing center. Figure 11 compares the estimated and actual mass of gas released over time, where the solid black line depicts the actual emissions through time (i.e., ground-truth emissions) and the dashed orange line shows the blindly estimated cumulative emissions. Over the duration of this single-blind, controlled release testing study, METEC released 700 kg of methane, while the estimated quantity shows a final mass of 673 kg, an underestimation by 27 kg, or about 4%.

The maximum difference between cumulative mass emitted and estimated mass emitted occurs on the 28th of August, when the estimated mass exceeded the actual mass by 38.5 kg. Over the course of 28 days, instances of short-term divergence between the two cumulative curves are observed, indicating inherent errors associated with the rate estimates for individual emissions. However, there are no signs of systematic bias toward overestimation or underestimation of emissions. For instance, an overestimation is observed during the first large emission event on August 27th, but the next couple of significant emission events are underestimated. These short-duration errors—when integrated over the entire testing period—largely cancel out, resulting in a relatively accurate estimation of the total mass emitted over the 28 days of the study.

While the site-level cumulative emissions estimate may be quite accurate, the source-level cumulative error may not necessarily be as reliable; considering Figure 5, there is some evidence of source misattribution, which may lead to suboptimal source-specific cumulative emissions estimates. To further investigate this, the cumulative emission estimate for each source group is compared to the ground-truth mass of emissions in Figure 12. In this figure, the solid lines depict the actual emitted mass, while the dashed lines show the estimated mass. In general, the estimates correlate with the overall trends observed in the actual emissions: an increase in the actual mass from a given group is often reflected as an uptick in the estimated quantity. The magnitude of these increases, however, does not consistently align with the actual mass emitted during an event, exhibiting both overestimations and underestimations (as demonstrated in Section 3.1). Additionally, source misattribution during individual emission events is evident in these curves. For example, the increase in mass associated with the 4S group (top panel) that occurs on September 13th is significantly underestimated. However, this underestimation is associated with a sudden increase in estimated emissions from the 4W source group at the exact same time, while the actual mass emission from this equipment group is nearly flat. In other words, during this emission event, the system is unable to distinguish between these two equipment groups and erroneously assigns a significant amount of emissions to an inactive source group.

Figure 13 shows the total mass emitted (black lines) as well as estimated quantities (colored bars) at the end of the testing (left panel), and the relative cumulative error (right panel) for each equipment group. This analysis shows that the cumulative emissions of the two highest-emitting equipment groups (4S and 4T) are underestimated, while the other three equipment groups show overestimated emissions. This highlights a systematic artifact of this particular system: a tendency to inflate the number of active emission sources by erroneously attributing a portion of the observed emissions to inactive sources, resulting in underestimation of the emission rates allocated to the equipment that is actually emitting. On average, these localization-related errors sum to a total site-level rate that is close to the actual site-level emission rate. While general trends and comparative analyses often yield reliable insight (e.g., the tanks emit more than the wellheads), the attribution of nonzero, secondary rates to individual emission sources should be interpreted within the context that the system tends to overproduce nonzero rates, especially if the attributed emission rate is smaller than the inferred rates from other equipment groups at the facility during a single emission event.

#### 3.3.1. Investigating Source Misattribution

As evident in Figure 12 and Figure 13, there are periods of time when the output of the system shows signs of source misattribution; it assigns nonzero rates to equipment groups that are not emitting during large emission events associated with a different piece of equipment. This section explores the underlying reason for source misattribution by analyzing high-resolution temporal data during two case studies: time periods exhibiting significant source misattribution. These two specific cases’ time periods are referred to as $t_{1}$ and $t_{2}$. Wind statistics, concentration measurements, and the correlation of source-sensor sensitivities are examined using a simple Gaussian Plume dispersion model. Considering Figure 12, there is a relatively large emission event from source group 4S on September 13th. During this event, a considerable portion of emissions were attributed to the 4W group, which was not emitting. This single event led to a considerable overestimation in the 4W group’s total mass estimate. A temporally refined view of this event encompassing a two-and-a-half-hour time interval ($t_{1}$) is illustrated in Figure 14. In this figure, the gray shaded region represents the extent of $t_{1}$. As shown, there were emissions from the 4W group directly preceding $t_{1}$, and the quantified estimate reflects this emission; however, the system then fails to appropriately identify the end of this distinct emission event, and the 4W group is erroneously assigned a nonzero rate for an extended period of time.

Another period of time ($t_{2}$) with significant source misattribution is shown in Figure 15. During this 1.5 h window, the dominant emitters’ (4S and 4T) rates are approximately constant, and there is a very small quantity of gas emitted from the 4W group. The estimated quantities, however, show significant emissions from 4 out of the 5 equipment groups (all but the 5W group are estimated to contribute significantly to the overall emissions during this time period). In the following analysis, this period of time is denoted $t_{2}$ and the previously described time window with significant source confusion (corresponding to Figure 14, Figure 15, Figure 16, Figure 17 and Figure 18) is denoted $t_{1}$.

Source-sensor alignment with respect to wind direction could be a primary reason for source misattribution. In order to investigate this, the specific layout of the facility (both sources and sensors) and wind statistics must be considered. The left panel of Figure 16 shows the potential source locations (circles), sensors (x’s) and the mean wind direction (black arrows) during $t_{1}$ and the associated concentration measurements in the right panel, where the colors of specific sensor locations correspond to the colors in the concentration measurements shown in the right panel. Figure 17 shows the wind rose plot associated with this same time period, $t_{1}$, indicating that the predominant wind during this period is originating from the north, with little variability in its direction. Considering the concentration measurements, wind rose, and relative positioning of the sensors with respect to the true emitter (the 4S group), it is evident that the SW sensor (pink) measures significant peaks in concentration due to the direct transport of pollutant from the 4S group that is directly upwind of it. During this time period, the wind direction is nearly constant, as shown in the wind rose. Due to the lack of wind variability, no other sensors indicate elevated concentration levels originating from the 4S group. Furthermore, the lack of variability in wind direction does not allow for emissions that potentially originated from 4W group to be detected by any other sensors, which could otherwise resolve emission source ambiguity between these two equipment groups. In other words, if the wind direction varied enough such that it, at some point during this period, pointed from the 4W group to a different sensor, and that sensor did not measure an enhancement in concentration, then this would result in strong evidence that the 4W group was not emitting, and this source ambiguity could be rectified.

In order to further demonstrate the underlying reason for these errors, we compute the so-called “source-sensor sensitivity matrix” using a simple Gaussian Plume forward dispersion model as described in [21]. This matrix represents the response of every sensor to every potential source, assuming a dimensionless emission rate of 1. Each row of this matrix represents a specific sensor at a given time, while the columns correspond to sources. The value associated with each element of the matrix represents the Gaussian Plume predicted concentration from the given source (column) at that sensor location at a given time (row). For illustrative purposes, only the SW sensor is considered, and only the source points from the 4S and 4W groups are used as potential source locations. Note that no other sensor is downwind from either the 4S or 4W group during the entire time period being considered. In other words, due to the low degree of wind variability, the predicted concentration on every other sensor is 0. As such, no other sensor provides information that may be used to better identify which group is contributing to the observed emissions. The time series of concentration predictions at the SW sensor for unit rate are shown in Figure 18. The correlation coefficient between signals corresponding to the two source groups is 0.85, indicating a high degree of similarity in the structure of these two predicted concentration curves.

Existing quantification algorithms generally assume a linear scaling of concentrations with rates (neglecting buoyant effects). As a result, they seek to fit predicted concentration profiles, such as shown in Figure 18 to the measurements shown Figure 16 via a set of linear equations $S\vec{Q}=\vec{b}$, where S is the sensitivity matrix, $\vec{Q}$ is a vector of source rates, and $\vec{b}$ represents the measured concentrations. In order to accurately attribute observed emissions to potential sources, the predicted concentration signals must be linearly independent. A high (near-one) correlation coefficient (as well as a simple visual inspection of similarity) between these two signals indicates that the predicted concentration time series are nearly identical to within an overall normalization factor. Therefore, the measured concentrations can be equivalently fit by an infinite number of combinations of these two sources. In other words, due to the lack of variability in wind direction and the fact that the only sensor receiving signals is directly downwind from two potential emission sources, there is not sufficient information during this time period to disambiguate between these two potential sources.

To mathematically demonstrate the fundamental degeneracy of this problem and explore how frequently these conditions occur, the condition number of the source-sensor sensitivity matrix during these time periods are computed. The condition number of S effectively represents how sensitive the inferred rates are to small changes (errors/noise) in the measurement vector. A higher number represents an “ill-conditioned” (i.e., less robust) inversion, while a lower number indicates a more stable and “well-conditioned” inversion. There are a multitude of factors that may impact the condition number of a sensitivity matrix associated with a given time period. Generally, the expectation with regards to variability of wind direction is that the condition number should be higher during periods of low wind variability, during which source misattribution may be more prevalent. This is because small changes in concentrations may result in a different inferred balance between distinct source groups, and because these groups are not necessarily close in space (i.e., both are upwind of the sensor, but at different distances), this can result in dramatic variations in the rate inferred from these different sources. In other words, low wind variability may result in ill-conditioned sensitivity matrices, which in turn leads to sub-optimal quantification estimates. In addition to computing the condition number associated with sensitivity matrices from $t_{1}$ and $t_{2}$, we also compute the circular standard deviation of wind direction ($\sigma_{\theta }$). Note that a low $\sigma_{\theta }$ indicates consistent wind direction with low variability, while large $\sigma_{\theta }$ corresponds to variable and turbulent wind patterns.

To determine the frequency of source misattribution-prone time periods, the condition number of the sensitivity matrix and the circular standard deviation of wind direction are computed over every three-hour time window during the 28 day testing period. The distributions of $\sigma_{\theta }$ and the condition number are shown in Figure 19. Here, vertical lines show the values associated with with the previously identified periods of significant source misattribution ($t_{1}$ and $t_{2}$). The percentile of these values with respect to the distribution from the entire testing period are shown in the legends. As shown in the left panel, the $\sigma_{\theta }$s during these manually identified time periods were 18.75 and 19.62 degrees, which fell in the 4th and 5th percentile of $\sigma_{\theta }$ values over the entire testing period, respectively. In other words, during these two time periods with some of the most significant source confusion in the estimated rates, the wind variability was unusually low compared to the rest of the 28-day study period. The condition numbers of the sensitivity matrices during these time periods were 4.8 and 4.35, which fell in the 83rd and 80th percentile of sensitivity matrix condition values computed over every three-hour window in the testing period. In other words, ∼80% of the testing period had sensitivity matrices with richer source-sensor pollutant transport information, which in theory results in more robust rate inversions compared to these instances.

In order to more quantitatively assess the impact of time periods with low amounts of wind variability across the entire 28 days of testing, each 15 min quantification estimate is classified as belonging to a period of low wind variability ($\sigma_{\theta }<25$)), or standard wind variability ($\sigma_{\theta }>25$); here, $\sigma_{\theta }$ is computed for a 3-h period centered on the 15 min quantification estimate. The covariance matrix of the group-specific errors for each of these sets of estimates is then computed. The off-diagonal components of these matrices represent the correlation of errors between two equipment groups’ quantification estimates. If an off-diagonal entry of this matrix is negative, then this indicates the tendency of the quantification errors for the associated equipment groups to be anti-correlated, i.e., as one equipment group’s rate is over-estimated, the other tends to be underestimated. As such, the off-diagonal components of these matrices can be used as a proxy for the tendency of two groups to be prone to source misattribution, with more negative values indicating a stronger tendency for source misattribution between the two groups.

Figure 20 shows the equipment group error covariance matrices for periods of time where $\sigma_{\theta }>25$ (left) and $\sigma_{\theta }<25$ (right), with the diagonal components (which simply represent the error variance of a given equipment group) set to 0 to highlight the dynamic range of the off-diagonal elements. It is evident that the periods of time with low wind variability tend to have significantly larger negative values in the off-diagonal elements, indicating a stronger tendency for source misattribution. The mean value of the off-diagonal components of the error covariance matrix for the period of low wind variability is −0.02, while the mean value for the “standard” wind variability periods is −0.003. In other words, the time periods with low wind-direction variability show a stronger tendency for source misattribution over the rest of the testing period by a factor of about 6.66.

Considering Figure 20, the equipment group combinations that show the most evidence of misattribution during low-variability conditions are the 4T-4S, 4T-4W, and 4W-4S combinations, which are all located on the west-most half of the facility. Note that the equipment groups that emit the most during the trial period are (by far) the 4S and 4T groups (see Figure 13), and that the dominant wind direction is out of the SE/SSE (see Figure 2). As such, the 4W equipment group is downwind of the largest emitters with respect to the dominant wind directions, and there is relatively sparse sensor coverage along the NW perimeter of the facility to help disambiguate between these three equipment groups along the dominant wind direction axes. The interplay of these underlying physical factors (dominant emitters, dominant wind direction, and source-sensor geometry) can explain the trends that are seen in misattribution between these sources. A more detailed and mathematical analysis of source misattribution and quantification accuracy with respect to sensor density and configuration is deferred to a future study.

These examples demonstrate that, even with dense sensor coverage, unfavorable conditions may occur that result in unavoidable source misattribution due to the inherent degeneracies between upwind sources associated with the geometry of plume dispersion during times of limited wind variability. While these conditions are relatively rare (these specific examples’ $\sigma_{\theta }$ were in the ∼5th percentile of the 28-day testing period), emission events occurring during these periods can negatively impact source-level emission quantification estimates. Several factors, including condition number and $\sigma_{\theta }$, can be employed to indicate uncertainties associated with emission rate estimates and perform quality checks for the periods involving ill-conditioned inversions.

### 3.4. Alerting of Anomalous Emissions Above Baseline

One of the primary applications of CMSs is the timely identification of anomalous emissions (i.e., “alerting”). As demonstrated in Section 3.1 and Section 3.2, instantaneous rate estimates are prone to significant noise, while time-averaged rate estimates show significantly less variance in the error distribution. For this reason, alerting based on instantaneous estimates is not recommended. In this section, we apply a rolling 12 h average to both the quantified rate estimates as well as the ground-truth rates, apply thresholds, and perform a binary scoring to assess whether the system was able to alert at a given threshold.

The controlled releases associated with this experiment included a preliminary week of “baseline” or “operational” emissions. As such, the goal of the system within the context of alerting of anomalous emissions is to identify “fugitive” events above this baseline. To this end, the average site-level rate is computed over the first week, and this value is subtracted off of the rolling-averaged rates, such that these quantities represent excess emissions over the baseline. A threshold of 0.4 kg/h is then applied, and a binary scoring is employed to identify each 15 min increment as either a “True Positive” (both the estimated and actual rates exceeded 0.4 kg/h over the baseline), a “False Positive” (the estimated quantity exceeded the threshold but the actual emissions did not), a “False Negative” (the estimated quantity did not exceed the threshold but the actual emissions did), or a “True Negative” (both estimated and actual emissions were below the threshold). The particular choices of employing a 12 h rolling average and a threshold above baseline of 0.4 kg/h for the alerting analysis are loosely motivated by the requirements for OOOOb Continuous Monitoring technologies, which are required to report at least one quantified rate every 12 h to be considered “continuous” and are also required to be able to detect a 0.4 kg/h increase above baseline (*continuous monitoring solutions must be able to determine a minimum leak threshold 0.4 kg/h over the site-specific background*, per the Environmental Protection Agency (EPA) [34]).

The black line in Figure 21 shows the actual site-level emission rate with a rolling 12 h average applied, while the blue line shows the rolling 12 h average of the estimated emissions. The mean baseline emissions for both curves have been subtracted off, and the threshold above baseline is depicted with a horizontal dashed black line. The gray shaded region highlights the first week of emissions that represent the “baseline period”. Green shaded regions indicate true positives, red indicates false negatives, and orange indicates false positives. If a time period is not labeled as any of these (i.e., the background is white), then it corresponds to a true negative time. The false positive rate (FPR), computed as $N_{FP}/\left(N_{FP}+N_{TN}\right)$, where $N_{FP}$ ($N_{TN}$) represents the number of 15 min increments identified as a false positive (true negative) and false negative rate, FNR, $N_{FN}/\left(N_{FN}+N_{TP}\right)$, are shown in the legend. For a 12 h rolling average and threshold of 0.4 kg/h above baseline, the corresponding FPR and FNR are 6% and 11%, respectively. Considering Figure 21, the majority of false negatives and false positives are not isolated events where the system erroneously produces a false positive or misses a large emission event. Rather, they are contiguously connected to periods of true positives. More specifically, many of the blocks of true positives are preceded by a brief false negative and have a brief false positive afterward. In other words, the output of the system has a delay in how it responds to a changing rate: when the rate increases, the 12 h rolling average quantification estimate does not increase immediately, resulting in a false negative on the leading edge of fugitive emission events (i.e., there is a finite time-to-detection), and similarly, on the trailing edge of the emission event, the estimated curve has a slight delay before it goes back below the threshold. There are, however, a few isolated false positives: time periods when the actual rate never exceeds the threshold, but some error in the quantification estimate results in the 12 h rolling average estimate exceeding the threshold. These cases, however, are short-lived and occur when the actual rate is close to (but does not quite exceed) the threshold.

In order to assess the time-to-detection (TTD) of the system in this alerting configuration (0.4 kg/h above baseline on a rolling 12 h averaging window), continuous periods in time where the actual 12 h rolling average emissions exceed the baseline by 0.4 kg/h are identified. Of these eight discrete emission events, seven of them were “detected”, meaning the estimated quantity exceeded the threshold at some point during that period, triggering an alert, and associated “true positive” in Figure 21. For these seven detected emission events, the difference in time between the first instance of a TP (i.e., the earliest time the system would “alert” on the given event) and the actual start of the emission event is computed as the TTD. Of these seven events, three were detected instantaneously (i.e., the estimated quantification showed an uptick above the baseline that was simultaneous to the actual rate increasing above that threshold), while the other four emission events had TTDs of 1.5, 2.25, 2.25, and 8.25 h, respectively, resulting in a total average TTD across these seven emission events of 2 h. The TTD will be influenced by many factors, including sensor density and configuration, source-sensor geometry, and the degree of wind variability. It should be noted that the quantification accuracy of the system also plays a role in the TTD, because the alerts are based off of quantification estimates exceeding a certain threshold.

## 4. Discussion

The crucial interplay between instantaneous error, improvements resulting from proper time averaging, and cumulative emission estimates are studied in this research. While short timescale, 15 min emission rate estimates exhibit higher instantaneous errors, this issue is partly mitigated and tends to be smoothed out when considering longer averaging periods. Therefore, cumulative mass emission estimates exhibit significantly lower error. This effect highlights the primary strength of CMSs for long-term use cases such as emissions inventory development, where the higher reliability of time-integrated data is crucial for a more accurate overall emission quantification.

Frequent source misattribution can negatively impact the confidence in the results of the CMS insights and cause inefficiencies in leak detection and repair efforts. As a result, it is crucial to minimize source misattribution by optimizing the CMS network configuration. In addition, quantification algorithm refinements are essential in achieving accurate and reliable emission rate estimates, and QA/QC processes and flags should be employed during periods of low wind variability when source misattribution may be most prevalent, if localized rate estimates are being used to inform decision-making.

This study, while offering valuable insights into the performance of the existing emissions quantification practices using fixed-point CMSs solutions, is subject to certain limitations. These limitations are primarily related to the short duration of the testing period (one week of baseline emission releases and then three weeks of baseline plus “fugitive” controlled releases). While insightful for initial analysis, the relatively short duration of the testing results implies that the entire range of atmospheric conditions that these systems are subject to in the field may not be present during this limited testing period. For instance, environmental factors such as wind speed and variability in direction, temperature, and atmospheric stability may exhibit seasonal variations that are not represented within this dataset. To best understand the deployment of these systems in the field, longer tests across more varied environmental conditions—ideally spanning multiple geographic regions that may exhibit different atmospheric characteristics—are needed. Nevertheless, some of the basic trends related to instantaneous error versus time-integrated error and the conditions that give rise to source misattribution should generally hold, although the specific values and details of the convergence of evaluative metrics with averaging time may be subject to change depending on other environmental and operational factors.

Limited variability in environmental conditions may limit the extrapolation of the quantitative error metrics from this study to a broader context. For example, the mean emission rate error and various metrics associated with the parity plots (Figure 4 and Figure 7) are derived based on the existing, limited dataset. Additionally, a longer data collection period can result in smaller uncertainty bounds associated with the error rates for various averaging times (Figure 8 and Figure 9). Similarly, our evaluation of the existing CMS’s capability to process varying emission rates is derived based on—and so, may be skewed by— the specific emission scenarios encountered during this limited-time controlled release testing study. A more comprehensive performance evaluation of CMSs can be done in the future using longer data collection studies under realistic operational scenarios that include a wider variety of emission patterns.

Another limitation of this study stems from the dense network of sensors employed during the trial period of this testing campaign. In typical field deployments, CMSs networks often contain between three and six sensors; however, for this blind test, there were ten sensors deployed. Generally speaking, technology providers tend to deploy a dense sensor network at these testing centers. This decision is largely driven by the inherent value of obtaining a large volume of high-quality, ground-truth data from controlled release studies. These datasets help drive innovation, and the more data that are collected during these testing campaigns, the better solutions are able to refine physical dispersion models and associated inversion frameworks to improve their technology. Furthermore, deploying a dense network allows for the post hoc investigation of system performance as a function of sensor density and configuration. In other words, the same quantification algorithm can be run across different subsets of the sensors to understand how the reliability of the output is dependent on the specific geometry of sources with respect to sensors and sensor density. The drawback of deploying a dense network for this study is that the results are not representative of field deployments and, as such, represent a best-case scenario in terms of DLQ accuracy. Therefore, these results need to be interpreted in this context. In general, the expectation is that source misattribution will become more prevalent with a smaller number of sensors because there are fewer sensors capable of breaking degeneracies between sources.

Section 3.3.1 investigates a limited selection of cases where there is a single primary factor contributing to source misattribution. As a result, straightforward conclusions can be made related to the root causes of the system performance degradation. However, in other cases, different factors may contribute to source misattribution.

While source-level emission rate quantification at high temporal resolution (e.g., 15 min estimates) offers a snapshot of emission rates at a specific moment, these estimates come with inherently higher uncertainties. As such, using only these short-duration estimates to derive insights may lead to a misunderstanding of true emission rates and result in wasted effort investigating what was only a spike in the noise of the emissions estimate. Note that in many applications, the objective of quantification is to either determine the contribution of different sources to the overall site-level emissions over an extended period of time, or to estimate the cumulative amount of emissions (or equivalently, average rate) over a long time period (e.g., a minimum of 4-h averaging). While mitigating the noise of the short-duration estimates would be helpful, due to some of the primary applications of CMSs, reducing the bias of the system’s quantification output is even more critical because it will result in more accurate quantification when longer-term averaging is considered.

The near-zero bias of the existing system is promising. It indicates that the appropriate selection of the averaging time window can improve the reliability of site-level emission rate estimates. As previously discussed, a short averaging window often fails to adequately mitigate the inherent noise in the short-duration estimates. On the contrary, an overly extended averaging window may reduce the amount of actionable insights present in the quantification signals. Therefore, the averaging time length matters: Figure 8 and Figure 9 suggest a steep improvement in the performance of the system by increasing the averaging time to 4 h. Increasing the averaging time from 4 to 12 h still results in some gains in the system’s performance. As the averaging time increases beyond 12 h, various metrics for the improvement of the system start to exhibit plateaued behaviors. Note that there is no right or wrong selection for averaging time selection. Instead, this decision should be informed by the objective of the methane measurement and quantification program. In some applications, the averaging time may be dictated by external factors (such as satisfying the requirement defined by the EPA OOOOb regulation), while other cases may present more flexibility.

Lastly, because fixed-point sensors rely on the wind to advect emissions from sources to sensors, the statistics of wind direction and relative positioning of sources and sensors plays a critical role in the performance of the system in terms of distinguishing emissions between different sources. As demonstrated, source misattribution may occur when there is little wind variability. This is an unavoidable limitation of these systems unless additional prior information or assumptions (e.g., operational data, independent measurements, aggressive sparsity promotion) are used to constrain or inform the emission estimates. As such, care should be taken when interpreting the equipment-specific emission rates in operational scenarios. In other words, the variability of wind direction and relative positioning of equipment and sensors should be considered before making decisions based on emissions estimates from point sensor networks to help mitigate the potential for equipment-specific “false positive” alerts.

Future research should focus on the evaluation of CMSs performance in more realistic emission environments. This includes longer and more rigorous testing under various meteorological conditions, covering a wider range of emission rates, and in the presence of complex baseline profiles. Longer controlled release studies will also help in better understanding the impact of various averaging strategies. Furthermore, ground-truth data on emission rates and concentration levels at given locations can be instrumental in comparative analysis of the performance evaluation of various emission quantification algorithms, including open-source and proprietary methods.

## 5. Conclusions

This study demonstrates that with a proper sensor network configuration, sufficiently accurate sensing hardware, and state-of-the-art quantification algorithm, a fixed-point CMSs platform can achieve low site-level cumulative error in long-term emission quantification under complex, asynchronous emission conditions, despite some source misattribution and high variance of short-duration error. This study employs data that were collected independently as part of a rigorous four-week single-blind controlled release study, conducted under operationally realistic and complex asynchronous emission conditions with fluctuating baselines and variable release rates from multiple sources. We strike a balance between capturing real-world emission complexities and maintaining analytical rigor through the systematic comparison of ground-truth data against blindly reported estimates across various averaging periods. This study explores the tradeoff between different ways of interpreting the quantified rates: noisy short-duration rate estimates that may be capable of capturing finer temporal insights, and longer timescale estimates that are shown to be prone to significantly less error, but may smooth over some relevant features. The findings of this study indicate a significant enhancement in emission-rate estimation accuracy by increasing the averaging time from 15 min to 4 h, followed by a more gradual improvement to an averaging time of 12 h, and diminishing returns beyond this. Note that this improvement comes at the cost of reduced temporal resolution, losing granular emission details. Therefore, an appropriate averaging time, balancing expected quantification error rates with the conservation of critical emission event details, is necessary.

The findings of this study are constrained by the short, four-week controlled release period, potentially under-representing the full spectrum of real-world atmospheric conditions and permutations of release source rates. The relatively dense sensor network (10 sensors vs. 3–6 in typical deployments) likely presents a favorable scenario for quantification accuracy, which may not be fully replicable in real-world applications. Finally, the reliance of these sensor networks on the wind to transport pollutant from sources to sensors introduces additional constraints, as low wind variability can lead to source misattribution, an inherent challenge for fixed-point CMSs platforms. Despite these acknowledged limitations, this study underscores that with informed decisions in the design of CMSs quantification algorithms, these systems can be a reliable source of information, specifically for long-term site-level emissions quantification, alerting of anomalous emissions above baseline, and informing other emissions measurement modalities. Looking ahead, future work will focus on enhancing the robustness and applicability of these systems through the evaluation of alternative inversion algorithms, the optimization of sensor placement strategies, and the rigorous testing of CMSs performance using data from longer single-blind controlled release studies as well as testing in the operational facilities.

## Figures and Tables

**Figure 1 sensors-25-02801-f001:**
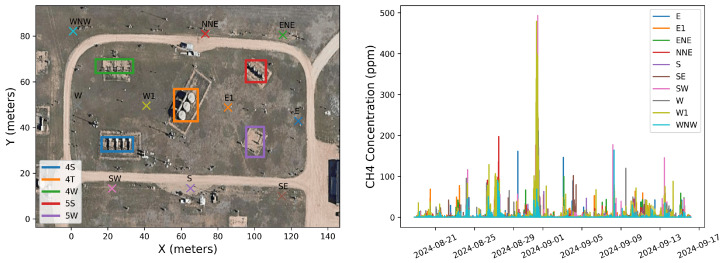
(**Left**) satellite image of METEC facility with sensor locations (x’s) and source groups (boxes) indicated. (**Right**) Concentration measurements over duration of 4-week data collection, with colors and positional labels corresponding to respective indications in left panel.

**Figure 2 sensors-25-02801-f002:**
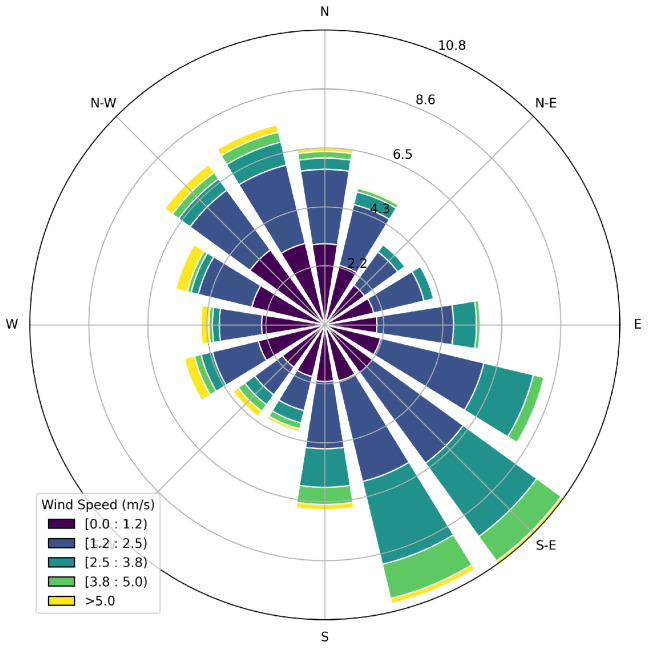
Wind rose from anemometer measurements during the entire 28-day testing period.

**Figure 3 sensors-25-02801-f003:**
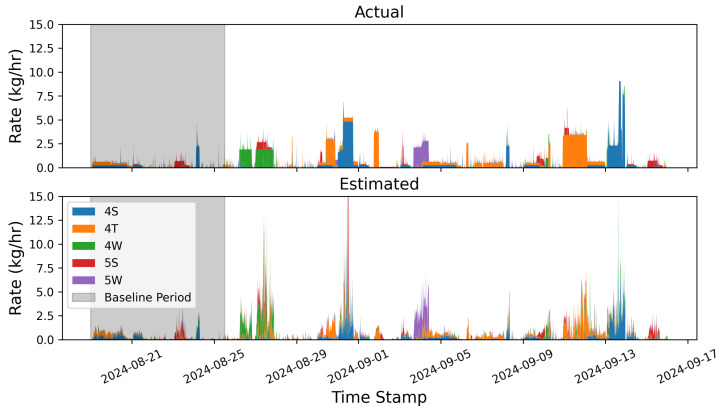
Stacked, overlapping bar charts showing the actual (estimated) emission rates in the top (bottom) panels, with each color corresponding to a source group. The duration of the baseline period is indicated with a shaded gray region.

**Figure 4 sensors-25-02801-f004:**
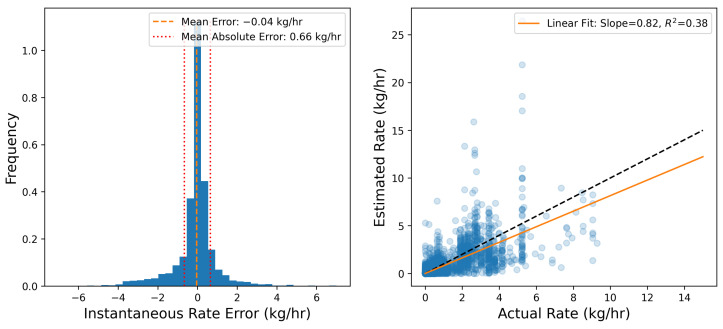
**Left panel**: Histogram of the 15 min emission rate estimate errors, with the mean error (−0.04 kg/h) and mean absolute error (±0.66 kg/h) indicated in vertical dashed lines. **Right panel**: Parity plot of the 15 min emission rate estimate compared to the actual emission rates, with the linear fit and the parity relation indicated with orange line and dashed black line, respectively.

**Figure 5 sensors-25-02801-f005:**
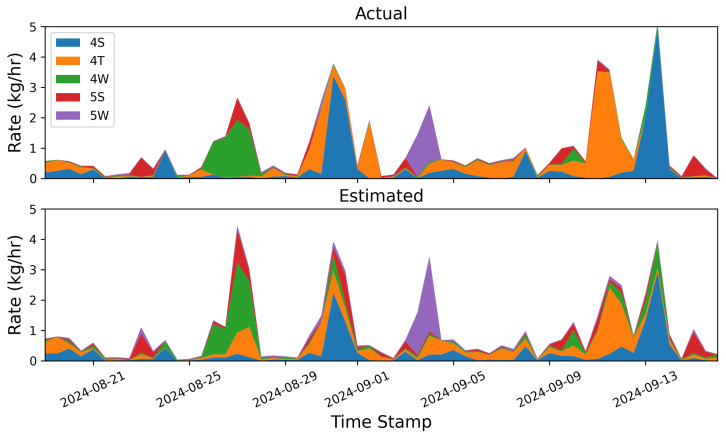
Actual (**top**) and estimated (**bottom**) 12 h aggregated rates.

**Figure 6 sensors-25-02801-f006:**
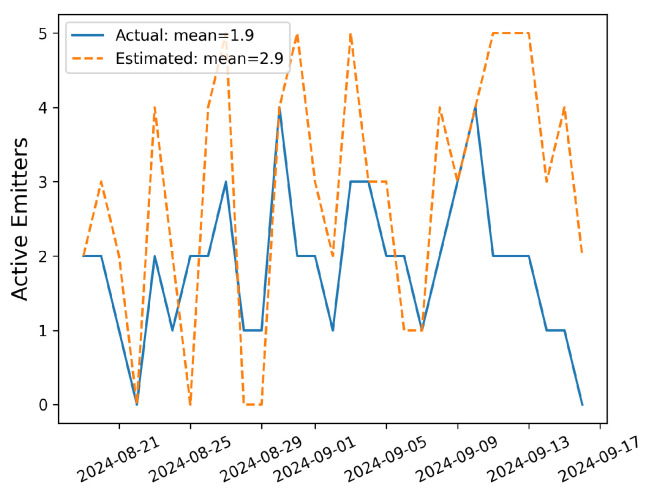
Number of active emitters as a function of time for 12 h aggregated rates.

**Figure 7 sensors-25-02801-f007:**
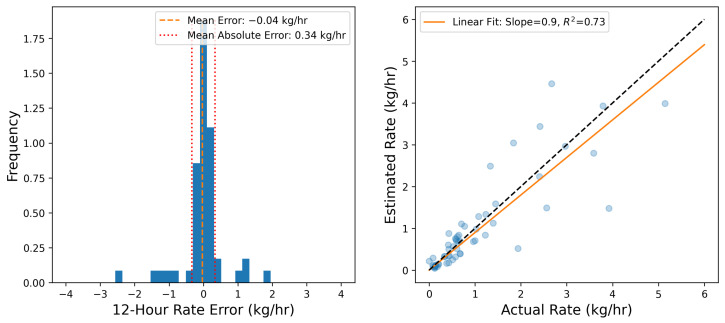
**Left panel**: Histogram of the 12 h emission rate estimate errors, with the mean error (−0.04 kg/h) and mean absolute error (±0.34 kg/h) indicated with vertical lines. **Right panel**: Parity plot of the 12 h emission rate estimate compared to the actual emission rates, with the linear fit and the parity relation indicated with orange line and dashed black line, respectively.

**Figure 8 sensors-25-02801-f008:**
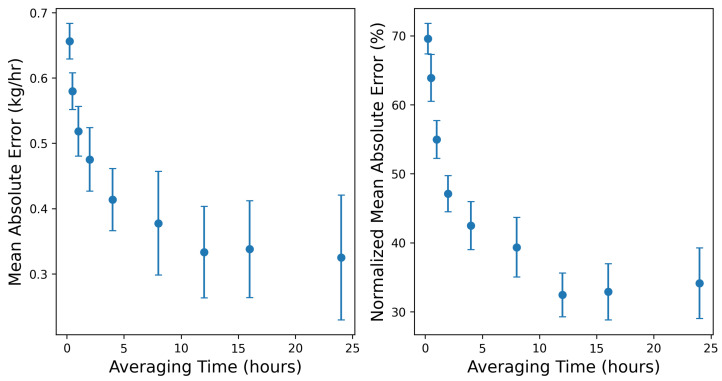
Mean absolute error (**left panel**) and normalized mean absolute error (**right panel**) as a function of averaging time.

**Figure 9 sensors-25-02801-f009:**
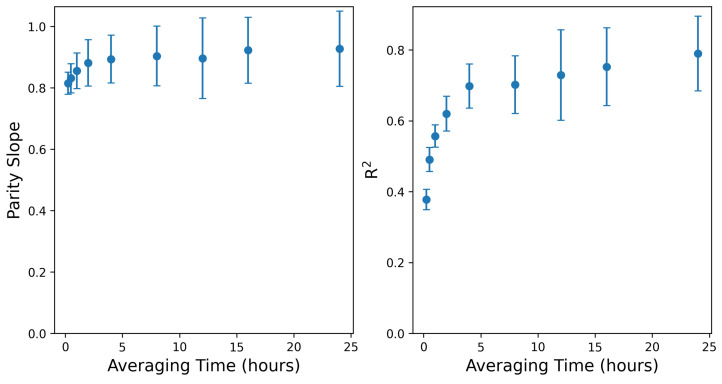
Slope of parity line (**left panel**) and associated $R^{2}$ (**right panel**) as a function of averaging time.

**Figure 10 sensors-25-02801-f010:**
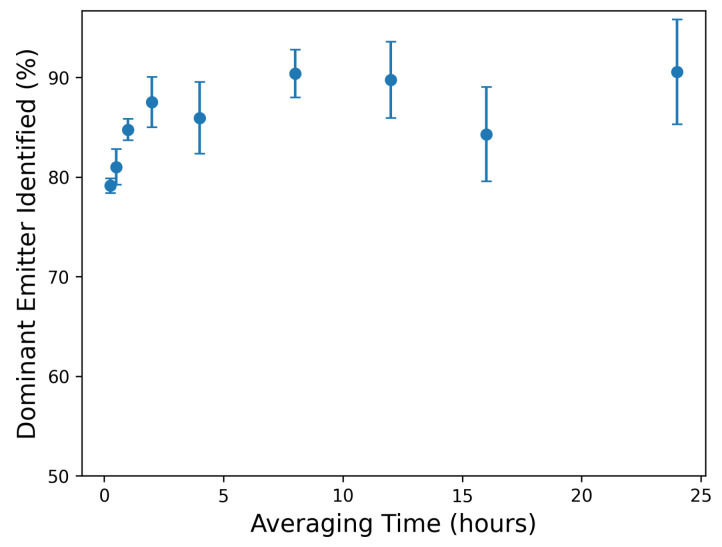
Correctly identified dominant emitter percentage as a function of averaging time.

**Figure 11 sensors-25-02801-f011:**
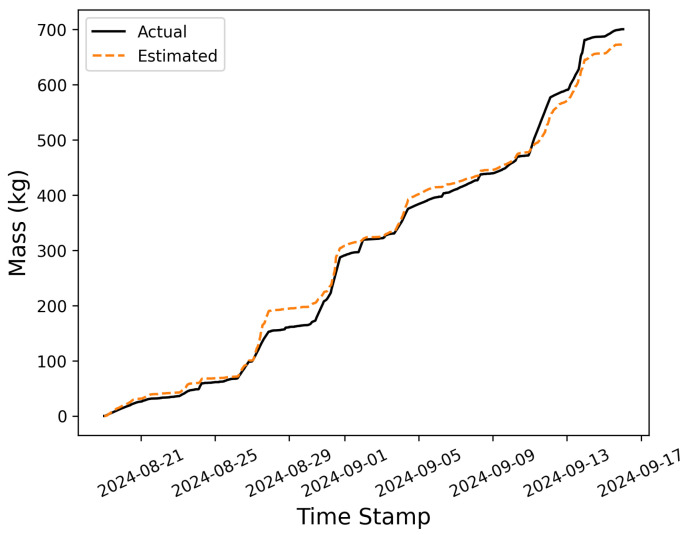
Cumulative emissions from the testing center (black), and cumulative estimated emissions (dashed orange).

**Figure 12 sensors-25-02801-f012:**
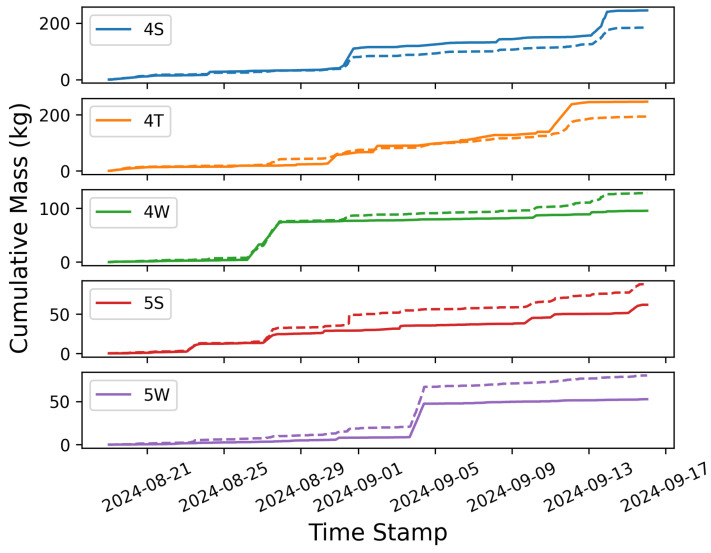
Cumulative emissions curves for each individual equipment group. Estimated quantities are shown with dashed lines, while true releases are depicted with solid lines.

**Figure 13 sensors-25-02801-f013:**
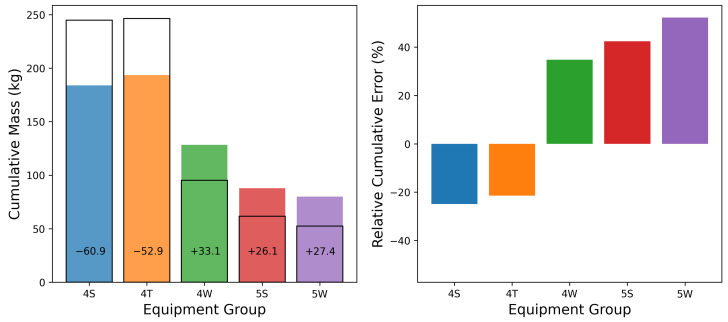
Equipment-group specific cumulative emissions (estimated and actual, **left**), and relative cumulative error (**right**).

**Figure 14 sensors-25-02801-f014:**
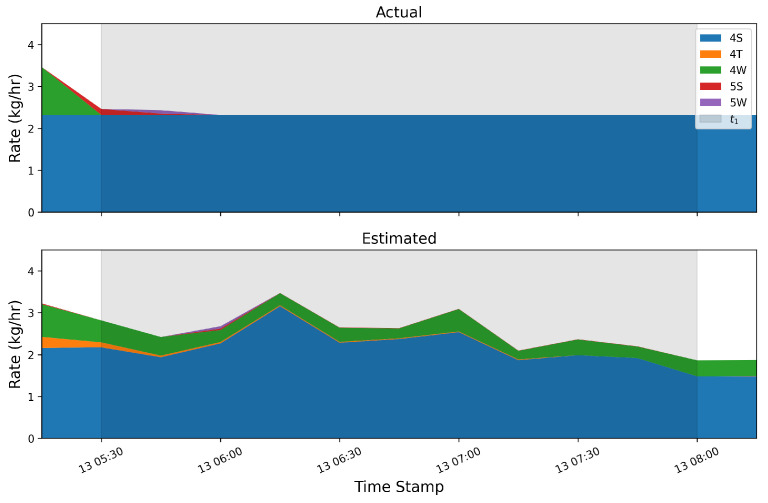
Actual (**top**) and estimated (**bottom**) emission rates by group during an emission event ($t_{1}$) from the 4S group that shows significant source misattribution in the estimated rates.

**Figure 15 sensors-25-02801-f015:**
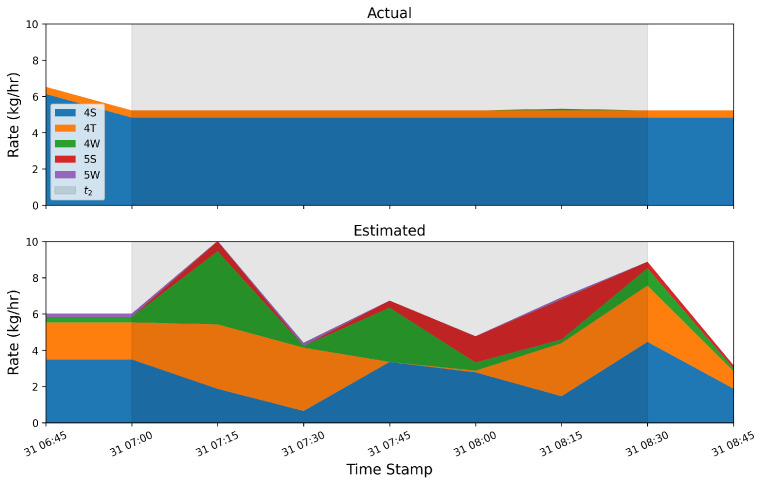
Period of source confusion, $t_{2}$, during which the estimated emissions are erroneously attributed to four distinct equipment groups, when only two of the groups are emitting significantly.

**Figure 16 sensors-25-02801-f016:**
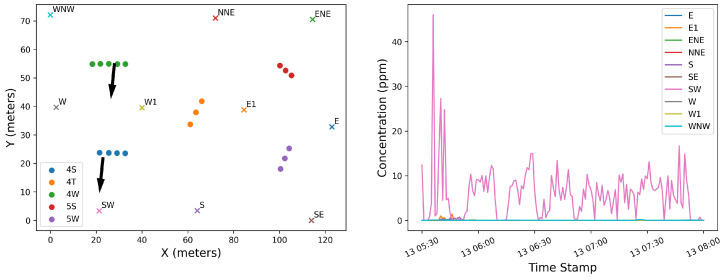
Layout of sources (dots), sensors (x’s), and mean wind direction (arrows) during time period with significant source misattribution ($t_{1}$). Only a single sensor captures elevated concentrations downwind of multiple potential source groups.

**Figure 17 sensors-25-02801-f017:**
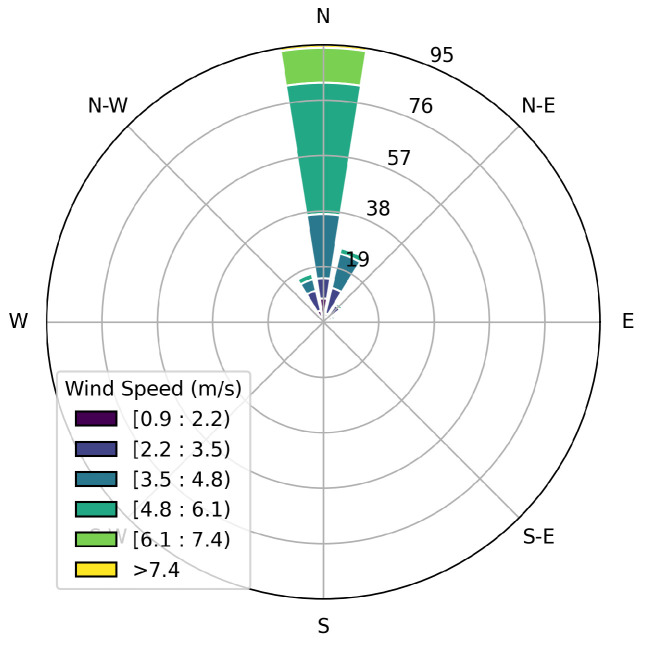
Wind rose during selected time period, demonstrating the small degree of wind direction variability.

**Figure 18 sensors-25-02801-f018:**
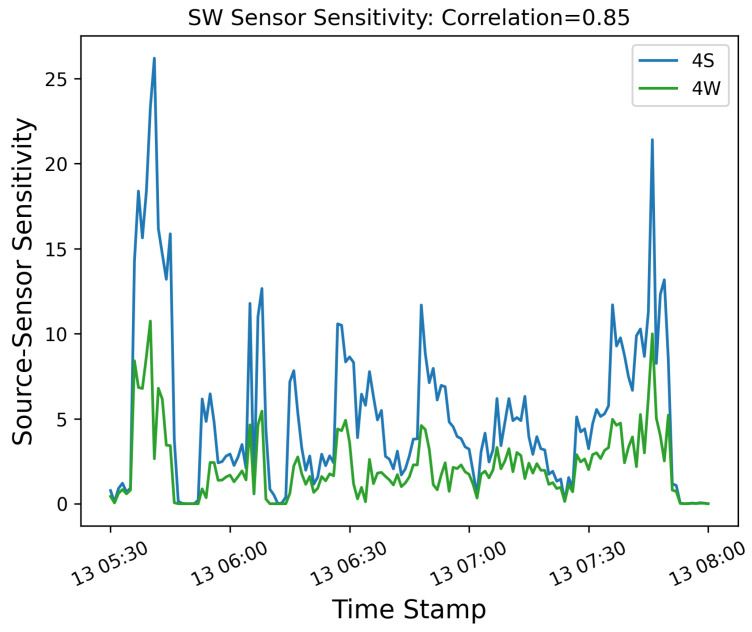
Source-sensor sensitivity for the 4S and 4W groups computed for the SW sensor. A high degree of correlation between the signals exists due to the specific geometry and uniform wind direction; because the sensor is directly downwind from both sources for the duration of the emission event, there is very little information that can help disambiguate between these two sources.

**Figure 19 sensors-25-02801-f019:**
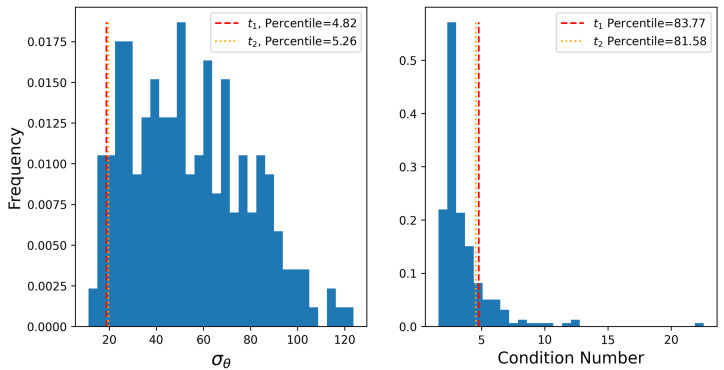
Histogram of circular standard deviation of wind direction (**left**) and condition number of sensitivity matrix (**right**) for every three-hour time window during the testing period. The respective values from the manually identified time periods ($t_{1}$ and $t_{2}$) with significant source misattribution are shown with dashed vertical lines.

**Figure 20 sensors-25-02801-f020:**
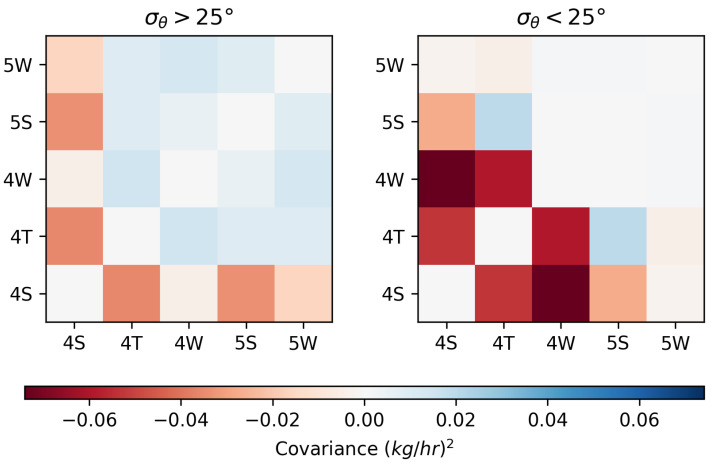
Covariance matrices with diagonal components set to 0 for periods of standard wind variability (**left**) and low wind variability (**right**).

**Figure 21 sensors-25-02801-f021:**
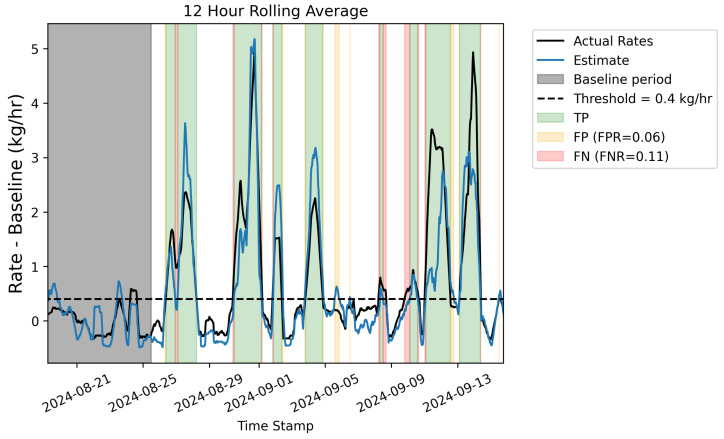
Twelve-hour rolling average of actual emission rate (black line), estimated rate (blue line). The black dashed line depicts a threshold of 0.4 kg/h over the mean baseline (the baseline period is shown with a gray shaded region), and the green, orange, and red shaded regions depict periods that correspond to true positives, false positives, and false negatives, respectively.

## Data Availability

Data for selected experiments is available upon request.

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
