# Peer review of "Performance Evaluation of Fixed-Point Continuous Monitoring Systems: Influence of Averaging Time in Complex Emission Environments"

_sensors, 2025, doi:10.3390/s25092801_

Round 1
Reviewer 1 Report
Comments and Suggestions for Authors
Summary:
This manuscript presents the results of a study designed to assess the performance of a fixed-point, continuous methane monitoring sensor network at a controlled emission release facility. The study builds on past work in this space by utilizing more complex methane emissions profiles – including time-varying ‘baseline’ as well as ‘fugitive’ emissions and multiple potential emission sources. Particular emphasis is placed on assessing the impact of time-averaging on the sensor network performance and includes additional subsequent analyses to further understand the results. Overall, this work is relatively comprehensive and the authors have done a good job of presenting the motivation for the work, laying out the study design, and thoroughly probing the results. However, there are a few places throughout the manuscript where further information, detail, and/or context is either needed or would further improve the readability of the work. This review is divided into two primary categories: major comments and minor comments. The major comments address aspects which I deem critical for publication (either through modification of the manuscript or providing sufficient explanation in a response); whereas the minor comments include more minor errors or small suggestions for improving the communication with the reader.
Major comments:
Section 1
The introduction ends with a statement about the insights presented from this study being relevant ‘for many analogous technologies’ despite the use of a specific sensor and associated data processing algorithms. This statement should be expanded in order to provide additional context about which insights will hold for other technologies – much of the analysis and results presented throughout the manuscript are actually directly tied to the sensor platform, configuration (both site and sensor), and the data processing. In fact, the Discussion section includes an acknowledgement that even with the same sensors different results would be expected if the sensor layout or density were to change (lines 646 – 651) and that extrapolation of specific metrics from this study to other cases could be limited (lines 622-623). It is critical in this case that the authors make the appropriate distinctions about which results/insights/findings can be extrapolated to other technologies and which are specific to this study.
Section 2
Site layout/configuration: The authors should include a description of the site and sensor layout in this section (Data and Method) rather than only providing a single high-level schematic in Fig. 12. Also, would it be possible to have this configuration figure include an overhead-style image rather than the marker-only schematic?
While the authors do include some discussion of the sensor layout for this study with respect to a more operational layout later in the manuscript (Section 4 Discussion), this information is critical to provide better context for the interpretation of the results and thus should be at least mentioned as part of the configuration discussion (requested above). Also, other sensor/site configuration information would be beneficial to include (e.g., heights for the equipment and sensors).
Sensor specifications (lines 201-202): Given that the sensor is laser-based, why are these specifications not provided in ppm*m? Additionally, what exactly is represented by the ‘sensitivity’ metric and/or how is this metric determined/calculated? If further information on the sensor is unavailable (e.g., proprietary from the vendor) or if it is available in a different publication then this should be stated.
Section 3
Underlying data: The authors should include high level stats (i.e., total number of 15-min intervals used; missing data; etc.) as part of the introductory discussion of the results.
Analysis method: Section 3.2 (line 346) – Can the authors update the text here to include more information on the bootstrapped resampling of the data and how that relates to the differing number of total samples for each averaging period (i.e., what are the 100 samples in each bootstrapped analysis; what is a ‘randomly-sampled realization’ of the data)?
Source misattribution: Section 3.3.1 (517-524) – The authors should consider introducing both case study time periods prior to conducting the source-receptor analysis that provides the explanation for the source misattribution. The current structure is a bit confusing since the second case study presents the same issues as demonstrated with the first case study and, yet, no additional insight from the source-receptor analysis is immediately included with the second case study.
The authors should more clearly present the relationship between the condition number (and subsequent sensitivity matrix) and the circular standard deviation of the wind direction (lines 525-549). It seems that with a fixed sensor–source configuration, the primary driver for a high condition number would be low variability in the wind direction (thus limiting the extractable measurement information in the inversion – as the authors have stated). I.e., these metrics are coupled and the standard deviation in the wind direction is the primary independent variable while the condition number is the dependent variable (and the metric more easily associated with the measurements).
Lastly, did the authors consider a re-analysis of some results after excluding time periods with low wind direction standard deviations (and corresponding high condition numbers) in order to assess the impact this (relatively small contribution) has on the overall interpretation of the results? (I am not necessarily advocating for this analysis – merely curious if it was considered or completed but not included in the manuscript)
Minor comments:
Lines 97-100: Reference [1] does attempt to disentangling sensor from data processing (with limitations) by processing sensor data from the three different platforms with a standalone data analysis package.
Section 2 (lines 214-232): The authors should mention the source misattribution analysis as part of the manuscript overview provided in this section.
Figure 2 (and possibly 4 for consistency): The authors should consider using semi-transparent markers for the scatter plot. The data density obscures the true metrics (as shown in the histogram) and emphasizes the outliers – better visual reconciliation between the panels will improve the messaging.
Line 286: missing closing parentheses
Figure 11: Can the authors expand x-axis just slightly to include before/after time period to provide more context around the time periods selected for investigation?
Section 3.3.1 (lines 452-463): The authors should include text indicating that two case studies will be discussed and already indicate time periods these cover as well as introduce the t1 and t2 designation (from line 522-523)
Author Response
Dear Editor and Reviewers,
We would like to express their sincere gratitude for your thorough review and valuable feedback on our manuscript. We appreciate the time and effort you have invested in volunteering to evaluate our work. We believe that your insightful comments and suggestions have been instrumental in improving the quality and clarity of our manuscript.
The attached document provides details on how we have addressed each of your comments (highlighted in plum), including revisions that we made to the manuscript. All the added and eliminated/reorganized text are highlighted in green and strikethrough, respectively.
We hope that our responses and the revised manuscript meet your expectations and address your concerns. Once again, we appreciate your constructive feedback and thank you for considering our manuscript for publication.
Sincerely,
Ali Lashgari, PhD
Manager of Scientific & Academic Partnerships

Reviewer 2 Report
Comments and Suggestions for Authors
Performance Evaluation of Fixed-Point Continuous Monitoring Systems: Influence of Averaging Time in Complex Emission Environments
Abstract
- Reorganize the abstract into 3–4 logical segments, each serving a distinct purpose. Begin with a succinct statement of the monitoring challenge in complex emission environments. Follow with a brief outline of the novel experimental design and quantification method. Then present the key quantified results. End with a clear statement of practical relevance or potential impact on future CMS deployments. Sentences should be shorter and more focused, using technical terms only where necessary and avoiding multiple embedded ideas in one sentence.
- Introduce the unique technical and scientific challenge within the first 2–3 lines. Emphasize that most prior validation efforts have involved simpler or more synchronized emission events, which do not reflect real-world conditions. State that this work is among the first to rigorously assess CMS performance under dynamic, overlapping emission scenarios with time-varying baselines. Position this clearly as the research gap being addressed, thereby increasing the perceived originality and scientific contribution.
- The term “single-blind controlled release study” appears early in the abstract but is never explained or contextualized. While this term may be well understood by a niche group of methane researchers, it risks alienating readers from adjacent disciplines or regulatory bodies unfamiliar with this protocol. Include a brief clarification directly in the abstract. State that in this testing design, methane release events were unknown to the CMS developers during the trial period, allowing for an objective performance assessment under realistic operating conditions.
- Introduction
- Introduce the research problem more clearly and much earlier. Position the work within the growing need for CMS to function accurately under realistic, complex emission conditions—such as asynchronous releases and fluctuating baselines—conditions underrepresented in previous testing protocols. Move this statement up within the first two paragraphs and contrast it briefly with simpler experimental paradigms used in prior evaluations. This adjustment will sharpen the focus and engage the reader early.
- Define “complex” emissions explicitly in the first half of the introduction. Detail the attributes—multi-source, overlapping emissions, time-varying release rates, and dynamic baselines—that distinguish these emissions from more controlled or static test environments.
- Provide a short comparative description of the evolution from METEC Phase 1 to Phase 2. Explain how the new protocol includes time-varying baselines and overlapping releases to better simulate operational emissions. This makes the paper’s contribution clearer, even to readers outside the immediate CMS field.
- While the technical discussion is solid, the introduction misses an opportunity to explain why CMS accuracy matters beyond academic interest—specifically in the context of emissions reporting, compliance, or field operations. Briefly connect the motivation for this study to real-world applications. Mention recent regulatory developments (e.g., EPA’s Subpart OOOOb) that require continuous emissions monitoring and establish expectations for system performance. This strengthens the rationale and shows alignment with industrial and policy relevance.
- Data and Method
- Include a tabulated or summarized breakdown of the controlled release parameters: total number of emission events, release rates (min, max, mean), duration range of events, and number of sources per event. This will allow readers to assess the realism and comprehensiveness of the test conditions and understand how they relate to operational scenarios in the oil and gas sector.
- Add a site schematic or aerial map showing the locations of all sources, sensors, and prevailing wind directions during the test period. Label equipment groups, include distance scales, and indicate dominant wind patterns. This visual context is essential for interpreting spatial relationships that affect detection sensitivity and inversion conditioning.
- Specify the native temporal resolution of the sensor output (e.g., 1-minute TDLAS readings), the frequency of meteorological data, and the precise process used for creating 15-minute and 12-hour average datasets. Explain whether overlapping or non-overlapping rolling averages were used. This transparency supports reproducibility and allows comparison with regulatory or operational reporting requirements.
- The paper states that the inversion framework was described in a prior study but provides only a cursory reference in this section. This prevents critical evaluation of the analytical process used to estimate emission rates. Summarize the key components of the quantification algorithm within this manuscript.
- Describe any data filtering or cleaning steps taken before analysis. Specify if any sensors were excluded due to malfunction, if background corrections were applied, or if signal smoothing was used. Mention any thresholds for invalidating data segments (e.g., due to wind stagnation) and how missing data were treated in the inversion routines.
- Explain why a range of averaging windows was selected and how these choices reflect trade-offs between temporal resolution and uncertainty reduction. If possible, cite previous studies or regulatory standards to support the use of each interval. Clarify whether these windows were chosen to evaluate alert responsiveness, inventory accuracy, or both.
- Results
3.1 Instantaneous Rate Error
- Clarify the basis for selecting 15-minute intervals as the standard resolution. Indicate whether this interval matches the native output frequency of the CMS algorithm, aligns with sensor sampling limits, or corresponds to external regulatory guidance. Explain if other output frequencies were evaluated during internal testing or if 15 minutes is widely adopted in field deployments.
- Include relative error metrics such as normalized mean absolute error (NMAE) or percent deviation from ground-truth values.
- Provide a brief explanation tying the observed error characteristics to physical causes. For example, mention whether short-term estimation errors are expected due to wind directional instability, sparse sensor coverage, or transient source confusion. If such behavior has been observed in similar CMS validation studies, cite those for context.
3.2 Time-Averaged Rate Error
- The section notes that the system tends to overproduce non-zero emission estimates for inactive sources, but this observation remains qualitative. There is no count, percentage, or mass quantification of misattribution. Quantify misattribution across all equipment groups. Report the percentage of estimated emissions assigned to inactive sources across the 12-hour averaging windows.
- Interpret what this higher R² means for field deployment. Is a value above 0.7 sufficient for regulatory compliance, emissions inventory confidence, or automated alerting? Compare this to known R² thresholds used in other sensor validation or atmospheric studies to anchor the result in a broader scientific context.
3.3 Cumulative Emissions Estimates
- Analyze spatial relationships between sensors and sources to determine if proximity or wind exposure explain systematic attribution errors. Identify whether certain areas of the facility were “blind spots” or prone to ambiguous wind conditions that hindered source differentiation.
3.4 Alerting of Anomalous Emissions above Baseline
- Explain the origin of the 0.4 kg/hr threshold—was it selected based on the distribution of baseline emissions, historical leak rates, or a minimum detection target from regulators? Consider including a sensitivity analysis showing how false positive and false negative rates change across different thresholds to support the selected value.
- Quantify alert latency for each emission event—how long after the emission onset was the alert triggered? Report average and maximum delays, and compare these across emission durations. Include a figure showing alert timing overlays to visualize detection alignment.
- Discussion
- Reorganize the discussion to synthesize how instantaneous error trends, time-averaged improvements, and cumulative estimates interrelate. Emphasize how source misattribution at short timescales is smoothed out in cumulative mass calculations, and what this means for CMS use cases ranging from real-time alerting to inventory development.
- Expand on how source misattribution affects key CMS applications. For example, discuss how frequent misattribution could undermine operator trust in CMS tools, delay repairs, or lead to inaccurate regulatory reporting. Mention how algorithm or network design improvements could mitigate this limitation.
- The manuscript positions itself as a significant evaluation of CMS under complex emissions, but does not compare its findings against benchmarks from other validation studies or platforms.
- Propose concrete next steps such as: testing under highly turbulent meteorological conditions; exploring hybrid averaging strategies for real-time vs. inventory applications; comparing algorithm types (e.g., Bayesian vs. deterministic inversions); and using mobile sensor systems to improve spatial attribution.
- Conclusions
- Begin with a statement that synthesizes the core achievement—demonstrating that a fixed-point CMS platform, when evaluated under complex, asynchronous emission conditions, can achieve low site-level cumulative error despite misattribution and rate variability. Emphasize the balance between operational realism and analytical rigor in this validation framework.
- Briefly acknowledge known limitations, such as misattribution during source overlap, difficulty detecting short-duration releases, or reduced inversion reliability under low wind conditions. This transparency reinforces trust and shows the authors understand the bounds of their claims.
- Conclude with a brief forward-looking statement. Highlight next steps such as testing alternative inversion algorithms, enhancing sensor placement strategies, or evaluating CMS in fully uncontrolled operational facilities.
Author Response
Dear Reviewer,
We would like to express their sincere gratitude for your thorough review and valuable feedback on our manuscript. We appreciate the time and effort you have invested in volunteering to evaluate our work. We believe that your insightful comments and suggestions have been instrumental in improving the quality and clarity of our manuscript.
The attached document provides details on how we have addressed each of your comments (highlighted in plum), including revisions that we made to the manuscript. All the added and eliminated/reorganized text are highlighted in green and strikethrough, respectively.
We hope that our responses and the revised manuscript meet your expectations and address your concerns. Once again, we appreciate your constructive feedback and thank you for considering our manuscript for publication.
Sincerely,
Ali Lashgari, PhD
Manager of Scientific & Academic Partnerships
